# The COX-2/PGE2 Response Pathway Upregulates Radioresistance in A549 Human Lung Cancer Cells through Radiation-Induced Bystander Signaling

**DOI:** 10.3390/biology12111368

**Published:** 2023-10-25

**Authors:** Alisa Kobayashi, Yota Hiroyama, Taisei Mamiya, Masakazu Oikawa, Teruaki Konishi

**Affiliations:** 1Single Cell Radiation Biology Team, National Institutes for Quantum Science and Technology, 4-9-1 Anagawa, Inageku, Chiba 263-8555, Japan; 2Radiation Effect Research Group, Department of Accelerator and Medical Physics, National Institutes for Quantum Science and Technology, 4-9-1 Anagawa, Inageku, Chiba 263-8555, Japan; 3Department of Radiological Technology, Graduate School of Health Sciences, Hirosaki University, 66-1 Hon-cho, Hirosaki-shi, Aomori 036-8564, Japan; 4Department of Radiology, Chiba University Hospital, 1-8-1 Inohana, Chuo-ku, Chiba 260-8677, Japan; 5Graduate School of Science, Rikkyo (St. Paul’s) University, 3-34-1 Nishi-Ikebukuro, Toshima-ku, Tokyo 171-8501, Japan; 6Electrostatic Accelerator Operation Section, Department of Accelerator and Medical Physics, National Institutes for Quantum Science and Technology, 4-9-1 Anagawa, Inageku, Chiba 263-8555, Japan

**Keywords:** radiation-induced bystander effect, radioresistance, COX-2, PGE2

## Abstract

**Simple Summary:**

The radiation-induced bystander effect (RIBE) is a phenomenon in which unirradiated cells respond to the effects of irradiation due to signals received from nearby irradiated cells. Experiments have shown that the RIBE can enhance cell radioresistance, which reduces the effectiveness of radiation cancer therapy. However, the RIBE mechanisms in vivo are still poorly understood. The methods employed in many in vitro studies on RIBE lacked direct contact between irradiated and non-irradiated cells; thus, they were insufficient to capture the full effects of RIBE in radiation cancer therapy. In vivo, cells are in contact with each other and perform intercellular responses through the gap junctions. In this study, we mimicked the RIBE in radiation cancer therapy in vitro by irradiating the subcellular region while maintaining cell-to-cell contact using a single-particle irradiation system to cell (SPICE-QST microbeam) facility at Chiba, Japan. Then, we investigated the contribution of the RIBE in the radioresistance of cancer cells and its mechanism focusing on cyclooxygenase-2 and its metabolite prostaglandin E2.

**Abstract:**

This study aimed to determine the mechanism underlying the modulation of radiosensitivity in cancer cells by the radiation-induced bystander effect (RIBE). We hypothesized that the RIBE mediates cyclooxygenase-2 (COX-2) and its metabolite prostaglandin E2 (PGE2) in elevating radioresistance in unirradiated cells. In this study, we used the SPICE-QST microbeam irradiation system to target 0.07–0.7% cells by 3.4-MeV proton microbeam in the cell culture sample, such that most cells in the dish became bystander cells. Twenty-four hours after irradiation, we observed COX-2 protein upregulation in microbeam-irradiated cells compared to that of controls. Additionally, 0.29% of the microbeam-irradiated cells exhibited increased cell survival and a reduced micronucleus rate against X-ray irradiation compared to that of non-microbeam irradiated cells. The radioresistance response was diminished in both cell groups with the hemichannel inhibitor and in COX-2-knockout cells under cell-to-cell contact and sparsely distributed conditions. The results indicate that the RIBE upregulates the cell radioresistance through COX-2/PGE2 intercellular responses, thereby contributing to issues, such as the risk of cancer recurrence.

## 1. Introduction

The radiation-induced bystander effect (RIBE), first described by Nagasawa and Little in 1992 [1], is thought to be an important phenomenon for radioprotection, particularly at low radiation doses. Furthermore, the mechanism behind the RIBE is the intercellular and tissue-to-tissue signaling that can be activated at higher radiation doses, such as those used in radiation cancer therapy [2,3]. The cancer microenvironment is complex and composed of multiple cell types [4]. One approach to understanding the mechanism of the RIBE in radiation cancer therapy is to investigate cellular signaling between irradiated cancer cells and nonirradiated cells [5].

To investigate this, researchers used specialized culture dishes to establish a gap between irradiated cancer cells and nonirradiated normal cells to allow the transmission of soluble signal molecules through the shared culture media [6,7]. However, this method was limited to assessing the effects of the medium-mediated intercellular communication (MMIC) pathway in the RIBE signaling pathway. Azzam’s group developed a better version of this technique to investigate not only the soluble factors of the RIBE but also the gap junction intercellular communication (GJIC) pathway [8,9,10]. They combined these techniques with inhibitors for either the GJIC or MMIC pathway. Regarding the MMIC pathway, they focused on the role of hemichannels, which are pores in the cell membrane that allow for the direct exchange of molecules between cells, which are part of the GJIC pathway and are open to the extracellular environment, to investigate how signaling events that induce DNA damage are transmitted to bystander cells [10]. Furthermore, the advancement of microbeam technology has accelerated RIBE studies, especially for mechanistic studies in the field of radiation therapy [11,12,13], due to its outstanding feature of targeting individual cells. For example, microbeams can be used to target only the cancer cells and exclude normal cells in a mixed cell population cultured in tissue-like monolayers, which is important for bidirectional signaling between cancer and normal cells. This makes the microbeam irradiation system a valuable tool for assessing the RIBE under conditions that mimic the in vivo environment of radiation cancer therapy. The advantageous features of microbeams enables studies to evaluate the RIBE between irradiated cancer cells and nonirradiated normal cells while preserving both the MMIC and GJIC pathways.

Notably, lung cancer is one of the most common cancers in men worldwide [14]. Radiation therapy, particularly particle beam therapy, has garnered attention as a treatment option for patients with other health conditions and a high risk of complications from surgery [14]. Many reported the overexpression of cyclooxygenase-2 (COX-2) in cancer that correlates with its radioresistance, which is also frequently observed in lung cancers [15]. As for RIBE, multiple molecules, such as nitric oxide, reactive oxygen species, and microRNAs, have been reported to act as bystander factors that promote radioresistance in human carcinoma cells [16,17,18,19]. Hei’s group found that cyclooxygenase-2 (COX-2) is a key player in the bystander signaling pathway, both in vitro and in vivo [10,20,21,22,23]. COX-2 and its metabolite prostaglandin E2 (PGE2) are regulatory enzymes and bioactive compounds involved in inflammatory, mitogenic, and angiogenic activities [24,25]. Previously, we reported the COX-2 expression and PGE2 production in both X-ray irradiated and co-cultured human lung cancer A549 cells [26]. Taking advantage of microbeams, we clearly show the promotion of DNA damage repair in irradiated A549 cells through the GJIC pathway [13]. In addition, there is much evidence that proves that bystander molecules contribute to cancer development, progression, and resistance [27,28]. Therefore, we hypothesized that RIBE-induced COX-2 expression is one of the reasons for radioresistance in cancer therapy. Concerning the experimental setup with microbeam irradiation, we reported earlier that A549 has normal connexin 43 with functional gap junction [13]. To elucidate whether RIBE is associated with bi-directional signaling between irradiated and non-irradiated bystander cells, we irradiated A549 cells that hold both functional GJIC and MMIC. Therefore, we believe that elucidating the effects and mechanisms of RIBE-induced radioresistance in the mimic of radiation cancer treatment for lung cancer, while preserving GJIC and focusing on COX-2 and its metabolite PGE2, carries significant societal importance. However, few studies have investigated the COX-2 and PGE2 responses in the RIBE under in vitro conditions that preserve both the MMIC and GJIC pathways.

In this study, we used a single-particle irradiation system to cell (SPICE-QST microbeam) system to induce the RIBE in vitro by irradiating a small region of cells in a culture dish while maintaining cell-to-cell contact. We investigated how the RIBE contributes to the radioresistance of human lung cancer cells by focusing on the role of COX-2 and its metabolite PGE2.

## 2. Materials and Methods

### 2.1. Cell Culture

We used three cell lines in this study: A549 human lung adenocarcinoma cells, A549-GFP cells that stably express GFP-tagged histone H2B fusion protein (H2B-GFP) [14,15], and A549-COX-2-KO cells (a human PTGS2 COX2-knockout A549 cell line). The A549 cell line was obtained from the RIKEN Bioresource Center (Ibaraki, Japan), and the A549-COX-2-KO cell line was purchased from Abcam (Cambridge, UK). The cells were cultured in Dulbecco’s Modified Eagle’s Medium (FUJIFILM Wako Pure Chemical Co., Osaka, Japan) supplemented with 10% fetal bovine serum and antibiotics (100 units/mL of penicillin and 100 µg/mL of streptomycin) at 37 °C in a 5% CO_2_ humidified atmosphere.

### 2.2. Microbeam Irradiation and Sample Preparation

All experiments with microbeam irradiation were performed using the SPICE-QST microbeam, which delivers 3.4 MeV protons (LET in water, 11.7 keV/μm) with a beam diameter of approximately 2 μm [11]. Within each point, 500 protons were delivered. The absorbed dose of A549 nuclei was estimated to be 5.2 Gy for 500 protons [12]. The fraction of cells targeted by a primary proton in the exposed populations was estimated to be 0.07–0.7%. After microbeam irradiation, the cells were incubated for 24 h before exposure to X-ray (Figure 1).

A day before irradiation, cells were seeded on a microbeam dish [11]. The cell dishes were prepared under two conditions: high-density conditions (HDC) and low-density conditions (LDC). For the HDC, A549-GFP cells were densely plated at 4 × 10^5^ cells/dish to promote cell-to-cell contact. We set the percentage of irradiated cells in the HDC cell dish to 0%, 0.07%, or 0.29%. The number of targeted positions was set to control the percentage of irradiated cells in an 8 × 8 mm area in the dish, which were irradiated in the X and Y directions in a matrix. For 0.07% and 0.29% irradiation (HDC 0.07% IR and HDC 0.29% IR), the number of points and the intervals for matrix coordinates were 27 × 27 (=729) points with 290 μm intervals and 54 × 54 (=2916) points with 145 μm intervals. To further investigate the involvement of the GJIC and MMIC pathways, we incubated A549-GFP cells under the HDC with a hemichannel inhibitor, 50 µM lanthanum chloride (La^3+^: Sigma-Aldrich, St. Louis, MO, USA) or sham 30 min before microbeam irradiation. The incubation continued until 24 h later, when X-rays were irradiated.

Under the LDC, cells were plated at 4 × 10^4^ cells/dish to be sparsely distributed to prevent cell-to-cell contact. For irradiation, a fluorescent image of the cells within an 8 × 8 mm area in the dish was obtained before microbeam irradiation. All cell nuclei were detected and their positions in the dish were calculated using the SPICE image analysis software (Microbeam version 3.4, YTHICK, Chiba, Japan). The X–Y coordinates of the cell nuclei were then output by the software. The desired number of X–Y coordinates to be targeted for irradiation was randomly chosen by the software. For the LDC dishes, 0, 75, or 750 cell nuclei were chosen for microbeam irradiation, which was calculated to be equivalent to 0% (LDC 0% IR), 0.07% (LDC 0.07% IR), and 0.7% (LDC 0.7% IR) irradiated cells in the cell dish. For experiments with A549-COX-2-KO cells, the number of cells and microbeam irradiation conditions were performed exactly the same with A549-GFP cells; however, experiments under the LDC were excluded.

### 2.3. X-ray Irradiation

For X-ray irradiation, an X-ray generator (TITAN, Shimazu Co., Kyoto, Japan) set at 200 kVp and 20 mA was employed, and irradiation was conducted through a copper and aluminum filter with a thickness of 0.5 mm, producing an effective energy of approximately 83 keV. The samples received X-ray at a dose rate of approximately 0.5 Gy/min.

To measure radioresistance by cell survival and micronuclei induction rate, the cells were exposed to X-ray at a dose of 4.4 Gy, which was determined as the 10% surviving dose (D_10_) from the survival curves obtained in a previous experiment. Details of the X-ray survival curves and the parameters calculated from the linear–quadratic (LQ) model equation are described in Appendix A.

### 2.4. PGE2 Treatment

To confirm the involvement of PGE2, a COX-2 metabolite, in radioresistance, we added PGE2 (Cayman Chemical, MI, USA) to A549 cells and examined its effect on COX-2 expression and radiosensitivity to X-rays. Specifically, A549 cells were seeded in a 6-well plate (BD Falcon, Bedford, MA, USA) at 2 × 10^5^ cells/2 mL/well and cultured for 1 day. The medium was then removed and replaced with fresh medium supplemented with PGE2 at concentrations of 0, 10, 50, 100, or 1000 pg/mL and incubated for 1 day. After incubation, the cells were subjected to protein extraction or X-ray irradiation.

### 2.5. Colony Formation Assay

Immediately after X-ray irradiation, the irradiated cells and controls were harvested using 2.5 g/L trypsin solution (Nacarai Tesque Inc., Kyoto, Japan) and plated in triplicate to obtain approximately 200 surviving cells per 10-cm dish (BD Falcon, MA, USA). The cells were incubated at 37 °C in a 5% CO_2_ humidified atmosphere. After 14–21 days, the cells were fixed with 5% formalin in phosphate-buffered saline (PBS) (FUJIFILM Wako Pure Chemical Co., Osaka Japan) and stained with 1% methylene blue (Sigma-Aldrich, MO, USA). Colonies that contained >50 cells were counted as survivors.

### 2.6. Micronucleus Formation Assay

Immediately after X-ray irradiation, 1 × 10^4^ cells (500 µL/well) were seeded in four-well chamber slides (SPL LIFE SCIENCES, Pocheon-si, Gyeonggi-do, Republic of Korea) and treated with 2-μg/mL cytochalasin B (Sigma-Aldrich, MO, USA). After 60 h of incubation at 37 °C in a 5% CO_2_ humidified atmosphere, the cells were rinsed with PBS and fixed in ethanol, and stained with 1 µM Hoechst 33342 (Dojindo Laboratories, Kumamoto, Japan) for nucleus and 2-µM Cell Tracker Orange (Thermo-Fisher Scientific, Waltham, MA, USA) for the cytoplasm. Fluorescent images were obtained using the SPICE-offline microscope system [11,14,15] and the Microbeam version 4.2 software (YTHICK, Chiba, Japan). The microscope was equipped with a 40× objective lens (UPlanFL N 40×, NA:0.75, Olympus Co., Tokyo, Japan) and a CMOS camera. A LED (X-Cite Xylis, Excelitas, Mississauga, ON, Canada) was used as a light source. In this study, we used distinct fluorescence mirror units for the two fluorescent dyes: U-MNUA2 for Hoechest 33342 and U-MNIGA3 for Cell Tracker Orange. Both mirror units were provided by Olympus Co., Japan. The wavelengths of each excitation/emission filter and the dichromatic mirrors were as follows: 360–370 nm/420–460 nm and 400 nm; 540–550 nm/575–625 nm and 570 nm. We obtained 525 images from one sample, consisting of 15 × 35 images captured in the X–Y direction, with individual image sizes of 166 × 166 μm. After capturing the images, we examined a minimum of 1000 cells, analyzing the number of micronuclei (MN) within binucleated cells (BN) and evaluating the fraction of MN per BN.

### 2.7. Protein Extraction and Western Blotting

Cells were collected in sodium dodecyl sulfate (SDS) (FUJIFILM Wako Pure Chemical Co., Osaka, Japan) sample buffer (4% (*w*/*v*) SDS, 125 mM Tris-HCl (pH 6.8)) and heated at 95 °C for 20 min. Each lysate (20 μg protein) was mixed with 4× loading buffer solution (FUJIFILM Wako Pure Chemical Co., Osaka, Japan), loaded into wells of 10% SDS-PAGE gels for electrophoresis, and then transferred onto polyvinylidene fluoride (PVDF) membranes (Merck Millipore Co., Darmstadt, Germany). After blocking in Tris-buffered saline containing 0.1% Tween 20 (Sigma-Aldrich, MO, USA) (TTBS) with 5% skim milk (FUJIFILM Wako Pure Chemical Co., Osaka, Japan), the PVDF membranes were incubated with the primary antibody at 4 °C overnight with gentle agitation. The antibodies for COX-2, glyceraldehyde 3-phosphate dehydrogenase, and horseradish peroxidase-linked secondary antibodies were purchased from Cell Signaling Technology (Beverly, MA, USA). All antibodies were diluted to 1/1000 in TTBS, and the membranes were washed twice with TTBS after every incubation step. The protein bands were visualized using a chemiluminescence substrate (Thermo-Fisher Scientific, MA, USA) and a chemiluminescence imager (ATTO, Tokyo, Japan). The concentration values of COX-2 and GAPDH were assessed using Multi Gauge software, Version 2.3 (FUJIFILM, Tokyo). The COX-2 expression level was calculated by dividing the COX-2 concentration value by the GAPDH concentration value. As a control, the COX-2/GAPDH value (COX-2 expression level) of the 0%microbeam irradiated sample was set to 1 for normalization, and the COX-2 expression level of the 0.07–0.7% microbeam-irradiated sample was calculated relative to this control.

## 3. Results

### 3.1. PGE2 Promotes COX-2 Expression and Cell Survival

First, we confirmed whether COX-2 and its metabolite PGE2 are involved in radioresistance. It has been reported that COX-2 is overexpressed in radioresistant malignant tumors [15]. PGE2, a metabolite of COX-2 is known to induce COX-2 expression through binding to E-series prostanoid receptors [24]. In our previous studies, the amount of PGE2 produced from irradiated A549 cells was approximately 100 pg/mL [26]. Therefore, we hypothesized that PGE2-mediated high expression of COX-2 causes radioresistance in cancer cells. To investigate the relationship between COX-2 expression and PGE2 concentration, 10, 50, 100, and 1000 pg/mL of PGE2 were added to the culture medium of A549 cells. After 24 h, COX-2 protein expression was analyzed using Western blotting. Figure 2A shows that COX-2 protein expression was higher in PGE2-treated samples than in untreated samples. Also, COX-2 protein expression increased with PGE2 concentration.

Furthermore, to investigate the effect of PGE2 on radioresistance, A549 cells were cultured in different concentrations of PGE2 at concentrations (0, 10, 50, 100, and 1000 pg/mL) for 24 h. Subsequently, the cells were irradiated with 5 Gy X-rays, and cell survival was determined using a colony formation assay. As a result, the survival rate was significantly increased in samples treated with 10, 50, and 100 pg/mL PGE2 compared with that in samples without PGE2 treatment (0 pg/mL). However, no significant difference was observed among the 10, 50, 100, and 1000 pg/mL concentrations (Figure 2B).

### 3.2. Microbeam Irradiation Promotes COX-2 Expression and Radioresistance in HDC Bystander Cells

COX-2 protein expression was analyzed by Western blotting after 24 h of exposure to HDC 0%, 0.07%, or 0.29% IR. As shown in Figure 3A, COX-2 protein expression was significantly increased by 1.89 folds in HDC 0.07% IR samples and by 1.64 folds in HDC 0.29% IR samples compared with that in HDC 0% IR samples. Next, we analyzed the radioresistance of microbeam-irradiated bystander cells. Twenty-four hours after microbeam irradiation, whole HDC cell culture dishes were exposed to 4.4 Gy (10% cell survival dose) of X-rays. Then, we determined cell survival and DNA damage with or without microbeam irradiation. Regarding the sensitivity to X-rays, for the 0% IR condition, the value was obtained from the sample that was not irradiated with anything (0% IR + with X-ray/0% IR + without X-ray). For the other samples (0.07–0.7% IR), we used the same percentage of microbeam-irradiated cells without X-ray irradiation as a control (0.07% IR–0.7% IR + with X-ray/0.07% IR–0.7% IR + without X-ray) (see Figure 3, Figure 4, Figure 5 and Figure 6).

As shown in Figure 3B, the survival of the HDC 0.07% IR was not significantly different from that of the samples that were not irradiated with a microbeam. In contrast, the HDC 0.29% IR sample exhibited a significant increase in survival compared to the without microbeam irradiated samples and 0.07% IR. As shown in Figure 3C, no significant difference in MN induction was observed between HDC 0.07% IR samples and HDC 0% samples. However, it was significantly suppressed by 0.69 fold in HDC 0.29% IR samples. Figure 4 shows the results of the LDC 0.07% IR and LDC 0.7% IR samples. Contrary to the HDC samples, the LDC samples did not show significant differences in COX-2 expression, survival rate, or MN induction rate. Furthermore, when HDC was supplemented with the hemichannel inhibitor La^3+^, there was no significant difference in X-ray survival between 0.29% IR and microbeam non-microbeam irradiated samples (see Figure 5).

Overall, the only condition that showed radioresistance against X-ray exposure was HDC 0.29% IR. From Figure 3, Figure 4 and Figure 5, we hypothesized that COX-2 and PGE2 responses are involved in radioresistance in bystander cells. To test this hypothesis, we used A549-COX-2-KO cells in HDC 0.29% IR. Twenty-four hours after microbeam irradiation, whole HDC A549-COX-2-KO cell sample dishes were exposed to 4.4 Gy of X-rays. After that, cell survival was determined using the colony formation assay, and DNA damage was assessed using the MN assay. The data on cell survival and MN rates are presented in Figure 6A and B, respectively. Interestingly, we did not observe X-ray radioresistance in A549-COX-2-KO cell samples.

## 4. Discussion

The RIBE causes cell death and chromosomal abnormalities in non-irradiated cells surrounding the irradiated area [1,27,28]. Irradiated cells also express COX-2, which synthesizes PGE2, a signaling molecule that promotes cell survival [29]. Chai et al. demonstrated the upregulation of COX-2 not only in the irradiated area restricted to a small region of 1 cm × 1 cm but also in the non-targeted regions of the tissue in X-ray irradiated mice [23]. Furthermore, prostaglandins are potent endogenous molecules that bind to E-series prostanoid receptors and act as trigger factors to activate COX-2 expression in cells [24]. Therefore, prostaglandins released from irradiated cells to neighboring bystander cells may induce the upregulation of COX-2 expression in cell populations, increasing radioresistance. RIBE-induced radio-adaptive response and rescue effect can be considered as the cellular response to radioresistance [27,28]. Based on the aforementioned reports, we hypothesized that the COX-2 expression and PGE2 production resulting from the RIBE affect the radiosensitivity of bystander cancer cells. The induction of radioresistance in bystander cancer cells may reduce the effectiveness of radiation cancer treatment.

To begin with, we confirmed the upregulation of COX-2 and radioresistance by PGE2. As reported by Kobayashi et al. [26], the PGE2 concentration in the media of cells irradiated with 5 Gy of X-rays was up to 100 pg/mL. As shown in Figure 2, PGE2 treatment at concentrations of 10, 50, 100, and 1000 pg/mL increased COX-2 protein expression in A549 cells, with a trend towards higher expression at higher PGE2 concentrations. However, there was no significant difference between the 100 pg/mL and 1000 pg/mL PGE2 concentrations. Furthermore, PGE2 treatment increased cell survival against X-ray irradiation, but there was no significant difference between the 10, 50, 100, and 1000 pg/mL concentrations (Figure 2B). These results suggest that PGE2 secreted by irradiated cells makes non-irradiated A549 cancer cells more resistant to radiation. However, it is also possible that the survival rate did not depend on PGE2 concentration due to the fast degradation of PGE2 [30].

Next, we measured the expression of COX-2 in microbeam-irradiated samples of HDC 0.07% IR and HDC 0.29% IR. As shown in Figure 3, both conditions showed a significant increase in COX-2 expression compared with the controls; however, no significant difference was observed between the two. In contrast, we found that HDC 0.29% IR increased cell survival and reduced the MN induction rate after X-ray irradiation compared with non-microbeam-irradiated cells, but not with HDC 0.07% IR. The non-radioresistance with a notable expression of COX-2 observed in 0.07% A549-HDC IR cells (Figure 3), can be explained that PGE2 is not the only signaling factor for upregulation of COX-2 expression and the increased radio-resistance is the result of activation multiple pathways via COX-2 expression. Indeed, others have reported that COX-2 and PGE2 are potentially involved in the induction of downstream cellular stress defense mechanisms. For example, PGE2 has been shown to induce the oxidative stress defense mechanism by HO-1 activation, which is a downstream gene of Nrf2 via COX-2/PGE2 [31,32]. Also, it is reported that COX-2 expression upregulates defensively against radiation-induced DNA damage and suppression of apoptosis through activation of multiple pathways such as STAT3, HIF-1α/PKM2 pathway [33,34]. However, the reduced cell survival with hemichannel inhibitor (Figure 5), which inhibited the PGE2 transmission from irradiated cells to the bystander cells, and the diminished radioresistance with A549 COX-2 knockout cells, which do not produce PGE2 (Figure 6), can be explained that PGE2 is the main signaling factor for radio-resistance through COX-2 expression. The difference between HDC 0.07%IR and HDC 0.29IR% is the number of cells irradiated in the cell dish, therefore, there will be a difference in the initial amount of PGE2 induced from microbeam irradiation between the two conditions. PGE2 is produced not only by the microbeam-irradiated cells but also by the near bystander cells that expressed COX-2, which amplifies the PGE2 signal.

As a result, we can assume that the difference in COX-2 expression induced by PGE2 in the microbeam irradiated cell dish between the HDC 0.07%IR and HDC 0.29%IR condition will be greater than the difference in the number of cells irradiated, due to the amplification by bystander cells. The amplification of PGE2 by bystander cells will occur later than the amplification in irradiated cells. The time course of the transmission and amplification process is unknown, but it is an important parameter in RIBE. Moreover, based on the reports of others [31,32,33,34] and our results, we considered that variations in downstream cellular defense mechanisms might arise depending on the initial levels of PGE2 production.

In addition, it should be noted that PGE2 is unstable and may have degraded before reaching nonirradiated cells from the microbeam-irradiated cells. Indeed, Okumura et al. [30] reported that PGE2 concentration in culture media could decrease to 50% within 20 min of incubation. Moreover, sufficient amounts of PGE2 would be necessary to upregulate the expression of COX-2 in bystander cells because of its fast degradation [35,36,37]. Furthermore, we did not observe any increase in COX-2 expression, cell survival rate, or MN rate with either LDC 0.07% IR or LDC 0.7% IR (Figure 4). The fast degradation of PGE2 may be the reason. Based on the result, we hypothesized that the RIBE is mediated by PGE2, which was transmitted to bystander cells through the GJIC pathway and exerted its bioactivity exclusively on neighboring cells. To verify the involvement of the GJIC pathway in HDC, we conducted experiments on HDC 0.29% IR cells cultured with La^3+^, a known hemichannel inhibitor [10] that inhibits the MMIC pathway. The results showed no significant difference in X-ray cell survival rate compared with that in the controls (Figure 5), which suggests that the density of the cells is critical in PGE2/COX-2 induced radioresistance.

To exclude the possibility of other pathways upregulating radioresistance in the microbeam-irradiated samples, we used the A549-COX2-KO cell line. Consequently, we observed that A549-COX-2-KO cells did not exhibit the radioresistance observed in A549-COX-2-wild-type cells. Since PGE2 is not produced in A549-COX-KO cells, these results support our hypothesis that COX-2/PGE2-mediated radioresistance occurs in bystander cells. Furthermore, the results may suggest that the amount of initial PGE2 produced by irradiated cells is responsible for the radioresistance in bystander cells, and the PGE2 produced by bystander cells in response to the initial PGE2. To clarify the underlying mechanisms, the biological response of bystander A549-COX-2-KO cells to irradiated A549-COX-2-wild type cells needs to be investigated.

Based on the results of HDC, LDC, and A549-COX2-KO cells (Figure 3, Figure 4, Figure 5, Figure 6 and Figure 7), we suggest that PGE2 transmitted through the MMIC pathway from irradiated cells is the dominant cause of the upregulation of COX-2 expression and radioresistance. However, this mechanism is likely only effective in bystander cells that are close enough to receive PGE2 before it degrades and can amplify the PGE2 signal.

Moreover, in vivo, cancer cells are known to create a microenvironment that promotes their own survival [4]. Microbeam technology is a powerful tool that can be used to mimic the tumor microenvironment and study RIBE between multiple types of cells. Further investigation is needed to understand the bi-directional signaling that upregulates the radioresistance in cancer cells through interaction with other cell types, such as immune cells, which may influence the RIBE through the alteration of different signaling pathways.

## 5. Conclusions

In this study, we mimicked the RIBE in radiation cancer therapy in vitro by subcellular region irradiation while maintaining cell-to-cell contact using a SPICE-QST microbeam, which emits proton beams indicating the vicinity of the Bragg peak in proton beam cancer therapy. We demonstrated that PGE2, a metabolite of COX-2, functions as a mediator of COX-2 expression. Furthermore, we identified the protein expression of COX-2 and the development of radioresistance in bystander cells. The induction of radioresistance was affected by the number of irradiated cells and cell density. Altogether, our findings suggest that radiation induces the bystander signaling-mediated COX-2/PGE2 response pathway that modulates radiosensitivity in cancer cells. These results may help determine the optimal target region including the tumor area that should be irradiated, to control these COX-2/PGE2 responses and enhance the efficacy of radiation therapy.

## Figures and Tables

**Figure 1 biology-12-01368-f001:**
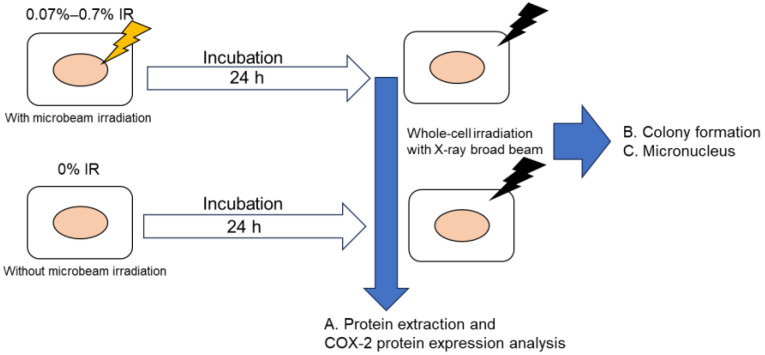
Timeline of the performed experiment on SPICE-QST proton microbeam irradiation followed by the X-ray irradiation.

**Figure 2 biology-12-01368-f002:**
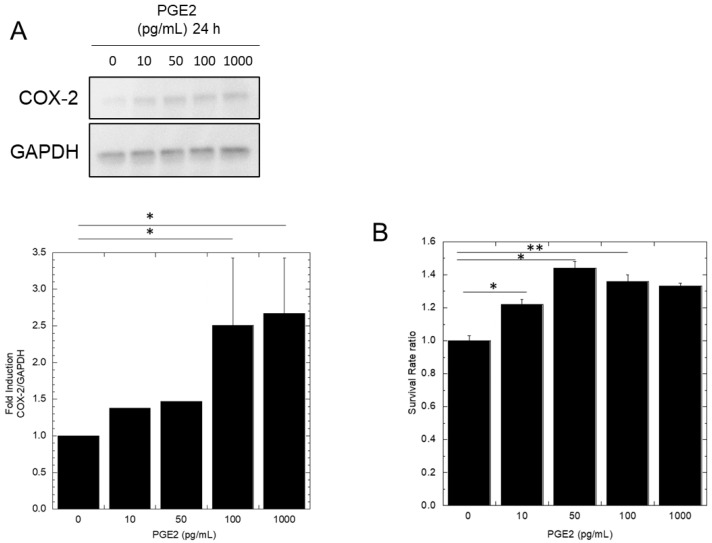
A549 cells (HDC) were treated with 10, 50, 100, and 1000 pg/mL of PGE2 for 24 h. Then, COX-2 protein expression was identified using Western blotting (**A**). A549 cells were treated with 10, 50, 100, and 1000 pg/mL of PGE2 or the vehicle for 24 h. Then, the cells were X-ray irradiated, and cell survival was determined using a colony formation assay. The survival fraction ratio represents the resistance rate against the without PGE2 sample (**B**). * *p*-value < 0.05. ** *p*-value < 0.01.

**Figure 3 biology-12-01368-f003:**
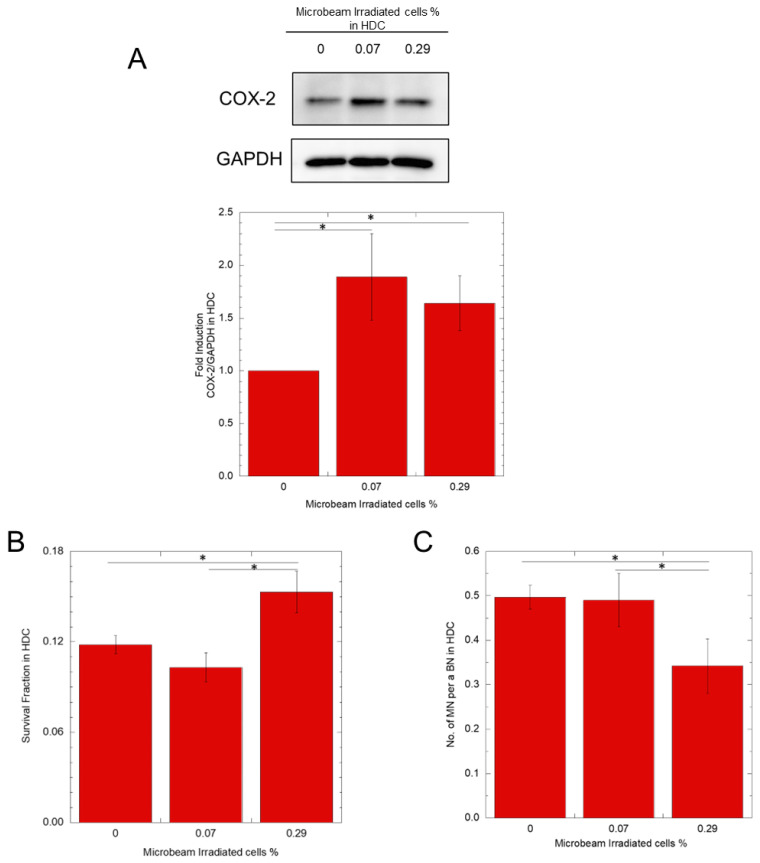
A COX-2 protein expression, survival fraction, and micronuclei induction rate of A549-GFP HDC samples after with or without microbeam irradiation. A total of 500 protons were irradiated per position, and the number of positions corresponds to the percentage of irradiated cells. Twenty-four hours later, Western blotting revealed an increase in COX-2 protein expression in the bystander cells (**A**). Regarding cell survival and micronuclei rate, cells were exposed to 4.4 Gy X-rays. Then, the 0.07% and 0.29% IR samples were compared against the without microbeam irradiated sample (0% IR) (**B**,**C**). * *p*-value < 0.05.

**Figure 4 biology-12-01368-f004:**
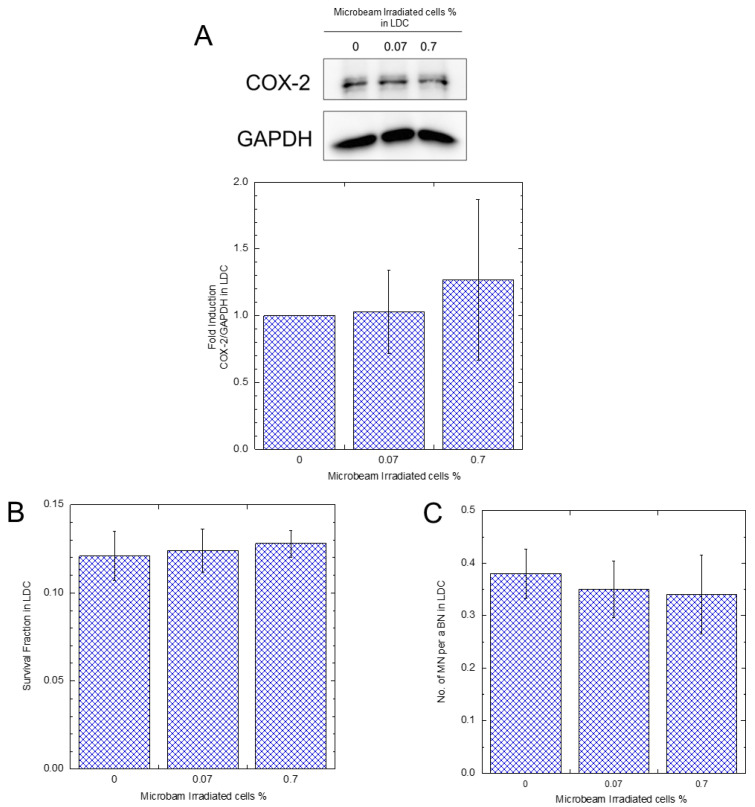
A COX-2 protein expression, survival fraction, and micronuclei induction rate of A549-GFP LDC samples after with or without microbeam irradiation. A total of 500 protons were irradiated per position, and the number of positions corresponds to the percentage of irradiated cells. Twenty-four hours later, Western blotting revealed the COX-2 protein expression in the bystander cells (**A**). Regarding cell survival and micronuclei rate, cells were exposed to 4.4 Gy X-rays. Then, the 0.07% and 0.7% IR samples were compared against the non-microbeam-irradiated sample (0% IR) (**B**,**C**).

**Figure 5 biology-12-01368-f005:**
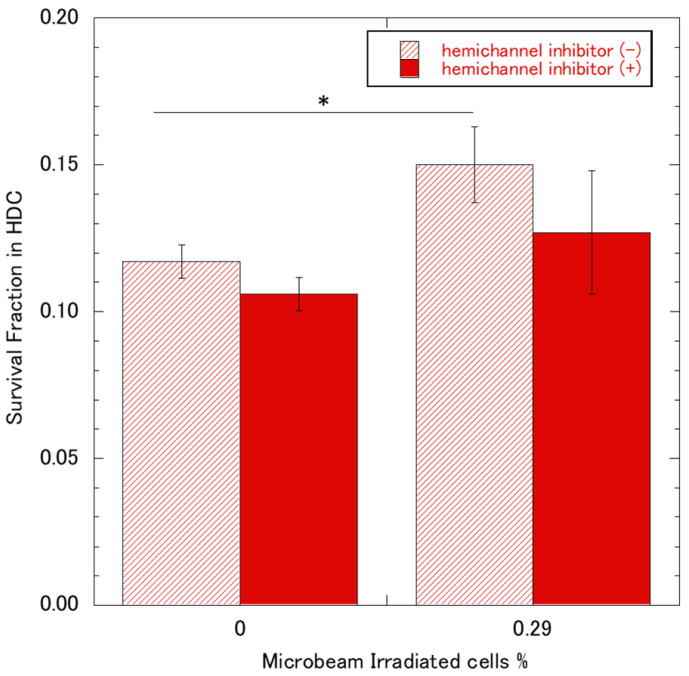
A survival fraction of A549-GFP HDC samples after 4.4 Gy X-ray exposure. A total of 500 protons were irradiated per position, and the number of positions corresponded to the percentage of irradiated cells. A549-GFP cells in HDC were incubated with a hemichannel inhibitor, 50 µM lanthanum chloride (solid bar), or sham (slant stripe bar). Twenty-four hours later, with or without microbeam-irradiated cells were X-ray irradiated, and the cell survival was detected using colony formation assay. * *p*-value < 0.05.

**Figure 6 biology-12-01368-f006:**
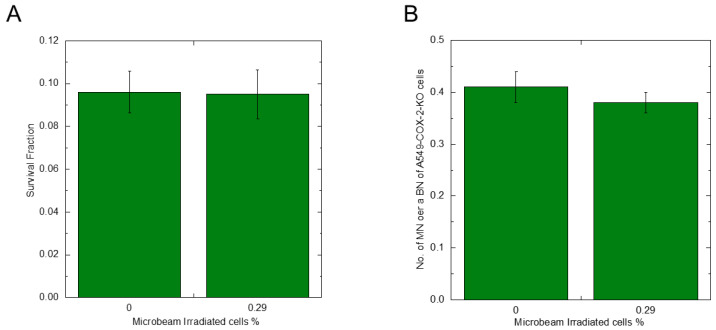
A survival fraction and micronuclei induction rate of A549-COX-2-KO HDC samples after 4.4 Gy X-ray exposure. A total of 500 protons were irradiated per position, and the number of positions corresponds to the percentage of irradiated cells. Twenty-four hours later, microbeam-irradiated and non-microbeam-irradiated cells were X-ray irradiated, and the cell survival and micronucleus rates were detected using colony formation and micronucleus formation assays (**A**,**B**).

**Figure 7 biology-12-01368-f007:**
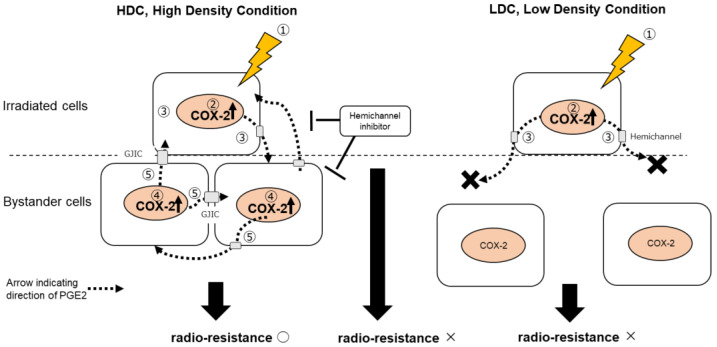
Schematic of the suggested model based on the main outcomes of this study. Responses occur in numerical order from 1 to 5. In microbeam-irradiated A549 cells (1), the expression of COX-2 protein (2) was induced and its metabolite PGE2 (3) was released into the extracellular environment. In HDC, bystander A549 cells were in close contact with each other, allowing PGE2 to act on neighboring bystander A549 cells and stimulate COX-2 expression (4). These COX-2-expressing cells continue to produce PGE2 (5), resulting in a feedback loop and amplification of the COX-2/PGE2 response within the cell population. Consequently, bystander A549 cells exhibit radioresistance. Hemichannels are involved as one of the propagation routes of PGE2, and inhibiting hemichannels diminished radioresistances. PGE2 may also be propagated through GJIC, which is composed of hemichannels. In LDC, in microbeam-irradiated A549 cells (1), COX-2 protein expression (2) was induced. However, the released PGE2 (3) cannot reach the surrounding bystander A549 cells and loses its activity. Thus, no propagation and amplification of the COX-2/PGE2 response were observed, and radioresistance was not induced in bystander A549 cells.

## Data Availability

Data supporting the findings of this study are available from the corresponding author upon reasonable request.

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
