# Peer review of "The COX-2/PGE2 Response Pathway Upregulates Radioresistance in A549 Human Lung Cancer Cells through Radiation-Induced Bystander Signaling"

_biology, 2023, doi:10.3390/biology12111368_

Round 1

Reviewer 1 Report

In the reviewed study authors aimed to investigate the radiationinduced bystander effect (called as RIBE) in lung cancer cells focusing on cyclooxygenase-2 (COX-2) and its metabolite prostaglandin E2 (PGE2). 

Authors mentioned that RIBE can trigger the radioresistance in cancer cells by signaling pathway. Is that a specific pathway for particular cancer cells (like lung cancer) or are there differences between them? It is not clear to me, why lung cancer cells were chosen in this study and previous?

Authors mentioned that within each point, 500 protons were delivered. Please add information about dose in Gy.

The samples received X-ray at a dose rate of approximately 0.5 Gy/min. What was the total delivered dose? From the results part, I suppose that 5 Gy, but it’s a relatively high dose. Could you comment on that?  

I am also, as a reader interested if only one dose was used. I suppose that mechanism might be different between low and high doses.

Cell were also exposed to 4.4 Gy (10% cell survival dose) of X-rays? How you calculate that 4.4 Gy?

In my opinion, the discussion part should be elaborated for example by adding data from different papers, and limitations of studies. I am not sure if the two last sentences are from that study or if this is a mistake. Please see "Authors should discuss the results and how they can be interpreted from the perspective of previous studies and of the working hypotheses. The findings and their implications should be discussed in the broadest...".

Reviewer 2 Report

In this article by using a single-particle irradiation system to cell (SPICE-QST microbeam), authors irradiated the subcellular region while maintaining cell-to-cell contact and show that radiation-induced bystander effect (RIBE) contributes towards radioresistance of cancer cells. The authors further show that the RIBE is mediated through an increase in the COX-2/PGE2 response pathway. The findings have important implications and may help in identifying new targets for radiation therapy (RT) of cancers as well as role of RIBE in RT outcome failures that currently is not well understood. However, there are a few concerns/suggestions:

1.    Based on the findings described in this manuscript, authors have concluded that PGE2, a metabolite of COX-2, functions as a mediator of COX-2 expression that mediates radioresistance through RIBE. However, this is a major concern since authors did not measure the levels of PGE2 at various times before or after microbeam irradiation or after X-ray irradiation. Without this, authors cannot conclude that PGE2 is a mediator of COX-2 expression since there is no direct evidence provided for this. This concern is further highlighted by the fact that at high density plating condition (HDC), there is an increase in the COX-2 expression yet radioresistance was not observed after microbeam irradiation of 0.07% cells. In fact, authors discuss (lines 333-337) that this could be due to the insufficient number of microbeam-irradiated cells because there will be insufficient production of PGE2 to induce radioresistance. However, this does not explain the reason for observed increase in COX-2 (since the above information suggests PGE2 is not involved) and why this increased COX-2 did not results in radioresistance. Therefore, measuring the levels of PGE2 and its correlation with COX-2 after irradiation is essential. (This also needs to be included in the discussion)

2.    Methods: 1) It is Important to clarify when the hemichannel inhibitor was added. Is it 30 min before microbeam irradiation or X-ray irradiation since this will significantly impact the interpretation of results. 2) As shown in the figures, it is not clear what does 0 microbeam irradiated cells (%) on X-axis represent. Please clearly describe in the methods that 0 means cells are not exposed to microbeam irradiation but are exposed to X-ray irradiation (so 0 means X-ray alone). Authors may want to include a schematic in Figure 2 for the irradiation schedule used. In addition, throughout the manuscript authors write, “microbeam-irradiated cells were X-ray irradiated” that conveys as if cells not irradiated with microbeam were not exposed to X-rays also- please re-phrase. Also clarify, if the untreated control without microbeam and X-ray irradiation was included in the experiments. If yes, did authors use this unirradiated control as a SF/PE of 1.0 to calculate SF for irradiated groups?

3.    Figure 1: 1) Since PGE2 is the metabolite of COX-2, it will be better that authors provide a rationale of investigating the effects of addition of PGE2 on the expression of COX-2. For example, add a statement here that it has been shown earlier that PGE2 affects expression of COX-2 through binding to E-series prostanoid receptors.  2) As can be seen from Figure 1B, there is no significant increase in SF at 1000 pg/mL of PGE2. Therefore, the expression of COX-2 should be shown at all the four concentrations of PGE2 in Fig. 1A (at least at 50, 100, and 1,000 pg/mL) to explain the reason of highest SF observed at 50 pg/mL and no significant change at 1,000 pg/mL (that is to demonstrate that if there is any correlation of SF with COX-2 expression). 3) Legend: Not clear if all the p-values shown are compared to 0 pg/mL PGE2 or with the other groups (with 10 pg/mL group, 50 pg/mL group etc.).

4.    Figure 3, Line 261: The statement is not correct since the increase is not statistically significant.

5.    Results from Figures 2-4 show that when cells are plated at high density then there is an increased COX-2 expression that corresponds with increased SF and reduced micronuclei, which is not the case when cells are plated at low density; and the addition of the inhibitor of hemichannel does not affect the SF (radioresistance). Based on these results, authors hypothesize that COX-2 and PGE2 responses are involved in radioresistance in bystander cells (lines 243-244). However, this is not a reasonable hypothesis based on the abovementioned results. The incorrectness of the hypothesis is further strengthened by the findings that despite an increase in COX-2 after microbeam irradiation of 0.07% cells (at high density), there is no radioresistance to X-rays in this group. Together, one may hypothesize that cell-to-cell interaction is more important for RBIE-mediated radioresistance that could be mediated by COX-2/PGE2. However, even this hypothesis is weak since as described above even at high cell density and with an increase in COX-2, radioresistance was not observed.

6.    Line 251, authors should mention whether the results disprove or prove their hypothesis (based on the comment 5 above, modify the rationale/hypothesis for the use of COX-2 KO cells).

7.    Figure 6: 1) Show here for HDC- how hemichannel inhibitor affects this mechanism. 2) There is a “?” between the two bystander cells (for HDC)- please mention in the figure legend what does this represent.

8.    What is the reason for maximum Y-scale value to be always 0.3 for SF, 1.0 for micronuclei formation, and 3 for COX-2 WB densitometry in all the figures? Please explain.

9.    Discussion: By using SPICE-QST and subcellular irradiation, authors are able to mimic the cell-to-cell interaction present in vivo and in the tumor microenvironment (TME). However, the TME consists of multiple cell types. The cells other than cancer cells such as immune cells would be affected differently by irradiation and may influence the RIBE through alteration of different signaling pathways. Authors may want to include this in the discussion as a limitation or future directions.

10.  Conclusions: Based on the findings in the manuscript, the authors cannot make a conclusion that these results will help in the development of personalized therapeutic approaches. With encouraging results observed after partial irradiation of tumors (only a part of the tumor is irradiated), the findings presented in this manuscript may help determine the optimal percent of tumor area that should be irradiated, depending on the type of radiation used. This can be added to the conclusions.

In addition, minor concerns and suggestions are:

1.    Add in the Western blotting methods how the densitometry for COX-2 expression was performed as shown in the bar graphs in the figures.

2.    The original images shown in the supplementary figures are not labeled. One cannot figure out what these images are showing.

3.    Line 232: Since the SF is not significantly changed, authors should not mention that it is increased.

4.    Lines 239-242: For clarity, please re-phrase this sentence.

5.    Line 255 and 261: Correct the typo for “revealed”.

6.    Part of the Figure 6 legend appears to be the part of the text of the results section. Please correct this.

7.    Please delete instructions from lines 352 to 355.

Round 2

Reviewer 1 Report

The authors substantially revised the manuscript according to suggestions.

Author Response

Dear Reviewer,

We would like to express our gratitude for taking the time to review our paper and for providing valuable comments and guidance.

Sincerely,

All authors

Reviewer 2 Report

Thanks for answering most of my concerns. The following remain unclear/unanswered:

1.    Comment 3 (2): Figure 1:  2) As can be seen from Figure 1B, there is no significant increase in SF at 1000 pg/mL of PGE2. Therefore, the expression of COX-2 should be shown at all the four concentrations of PGE2 in Fig. 1A (at least at 50, 100, and 1,000 pg/mL) to explain the reason of highest SF observed at 50 pg/mL and no significant change at 1,000 pg/mL (that is to demonstrate that if there is any correlation of SF with COX-2 expression).

Authors have answered that in a previous study they measured the concentration of PGE2 produced after irradiating A549 cells with various doses of X-rays. As a result, authors measured a maximum of 100 pg/mL of PGE2 and accordingly selected several concentrations (centered around 100 pg/mL) of PGE2.

However, I was suggesting that expression of COX-2 should be shown at all the four concentrations of PGE2 in Figure 1A (now it is shown at 100 pg/mL only), not the concentration of PGE2 after irradiating A549 cells with various doses of X-rays (please see my previous comment). Accordingly, the statement added to the discussion by the authors needs to be modified.

2.    Comments 4: Figure 3, Line 261: The statement is not correct since the increase is not statistically significant.

This statement was for the expression of COX-2 in the Figure 3 legend (now Figure 4, line 290). Please correct this.

3.    Comments 8: What is the reason for maximum Y-scale value to be always 0.3 for SF, 1.0 for micronuclei formation, and 3 for COX-2 WB densitometry in all the figures? Please explain.

Authors replied that they designed it to comfortably accommodate error bars, significance bars, and annotations, ensuring that the bar graph displaying numerical value is as visually accessible as possible.

I do not agree with this response. In fact, SF is less than 0.2, MN is less than 0.6, and COX-2 expression less than 2.5, even after considering error bars in all the figures. If the Y-scale is reduced accordingly, differences between treatment groups will be better visualized for comparison.

4.    With regard to my earlier comment about the original blots and gels not annotated (comment 11), I still did not find the files that show labeling/annotations.

5.    Comments 12: Line 232: Since the SF is not significantly changed, authors should not mention that it is increased.

Authors responded that Figure 1 has been revised to show that the PGE2 addition conditions of 10, 50, and 100 pg/mL have significantly higher SF than 0 pg/mL.

However, this comment was for figure 2B, line 232 (now Figure 3B, line 254). Two statistical comparisons are shown in the figure: 0 vs 0.29% and 0.07% vs 0.29% and that is what should be described in the results.

6.    Comments 13: Lines 239-242: For clarity, please re-phrase this sentence.

Similar to the above comment 12, authors have not corrected the sentence that was mentioned on lines 239-242 (now lines 261-264). “Furthermore, in HDC with the addition of a hemichannel inhibitor La3+, even when 0.29% IR was applied, the increase in cell survival observed under the condition without the addition of a hemichannel inhibitor was not shown (Fig. 5)”.

New comments:

1.    Change on line 265: From Figures “3-5” (as the figure numbers are changed now) and please check the figure numbering referred throughout the manuscript.

2.    Add “emits” on line 421: …. which proton beams indicating the vicinity of the Bragg peak in proton beam…

There are multiple sentences that have words missing or have grammatical and syntax errors. Some need to be rephrased. Especially, the new text added to the manuscript after first revision needs extensive editing for English language.
